# Double Trapezoidal Wave Transmitting System with Controllable Turn-Off Edge

**Yuan Jiang [1,2], Yanju Ji [1], Yibing Yu [1], Shipeng Wang [1] and Yuan Wang [1,*]**

[1]   College of Instrumentation and Electrical Engineering, Jilin University, Changchun 130026, China;
      jiangyuan0816@163.com (Y.J.); jiyj@jlu.edu.cn (Y.J.); yuyb19@mails.jlu.edu.cn (Y.Y.);
      wangsp19@mails.jlu.edu.cn (S.W.)

[2]   Changchun Institute of Optics, Fine Mechanics and Physics, Chinese Academy of Sciences,
      Changchun 130033, China

[*]   Correspondence: wangyuan_ciee@jlu.edu.cn; Tel.: +86-135-0432-1424

**Abstract:** For time domain transient electromagnetic measurement, the negative sign often appears in the polarization region, which contains the induced polarization information. It is considered that the polarization effect is caused by the capacitance charge of the earth. Extending the turn-off time of the emission current means increasing the charging time, and reducing the charging voltage, which makes the polarization effect easier to observe. Therefore, a double trapezoidal wave transmitting system with a controllable turn-off edge is designed in this paper. In the process of current transmitting, the turn-off time can be controlled by changing the clamping voltage depending on the passive clamping technology. By cutting into the absorption resistance, the current oscillation can be eliminated under the condition of ensuring linearity. To verify the effectiveness of the system, we designed a polarized wire loop based on the filament model simulating the polarized earth. Comparing the response of the wire loop, the emission current with short and long turn-off times contributes to inducing the induction and polarization fields respectively. The double trapezoidal wave transmitting system with a controllable turn-off edge is suitable for measuring the induced polarization effect.

**Keywords:** induced polarization effect; transmitting system; turn off time; polarized wire loop

## 1. Introduction

Time domain transient electromagnetic method (TEM) is a common mineral exploration method. It is widely used because of its high lateral resolution, sensitive response to low resistivity, and the ability to distinguish small underground heterogeneous bodies. In the measurement of the nonpolarized region, the transient electromagnetic signal is approximately a monotone attenuation e-exponential curve that contains the resistivity information of the earth [1,2]. However, a negative sign often appears in the transient electromagnetic signal measured in the polarization region. In 1980, Spies first interpreted the negative transient electromagnetic response of the data in TEM [3]. In 1982, Weidelt theoretically proved that if the earth's resistivity does not have dispersion characteristics, the transient electromagnetic response during the off-time after excitation will not change sign at any time, which stimulates the study of the influence of induced polarization characteristics of the earth on the transient electromagnetic response [4]. The above research showed that the induced polarization effect is the key to the negative sign in TEM. Recently, more studies focused on the induced polarization effect in TEM measurement. Kozhevnikov discussed the influence of the induced polarization effect on TEM inversion through numerical simulation experiments [5]. Kang developed a procedure to invert time domain induced polarization (IP) data for inductive sources [6].

The Cole–Cole model proposed by Pelton is considered to be an effective equivalent model for describing the induced polarization effect of the earth, and the extended version, i.e., the filament model, proved useful in the analysis of physical processes [7]. The circuit of the filament model is shown in Figure 1, where the energy storage lies in a simple capacitor.

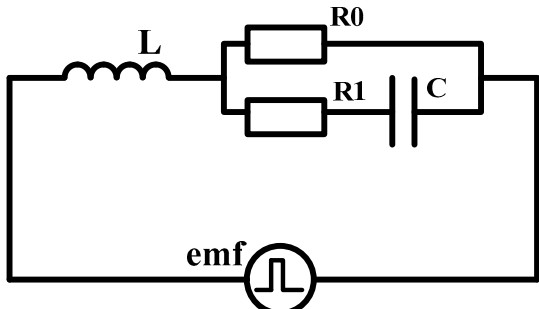

**Figure 1.** The circuit of the filament model.

Kozhevnikov discussed the excitation of transient induced polarization responses using current and voltage sources and proposed that the induced polarization effect of the earth should be observed using an ungrounded loop [8]. Du reported TEM measurements employing a LT SQUID system as a B-sensor in a polarizable and resistive survey area and proposed that the flux changes imposed in the ground by switching off the primary field in TEM can be considered as a pure voltage source (emf); the sign reversal in TEM is due to the discharging of in-ground capacitance (C) [9]. The turn-off edge of an emission current in an ungrounded loop directly affects the measurement result of the induced polarization effect.

The conventional transmitting system of TEM measurement is currently used to measure the induced polarization effect [10,11]. TEM is based on step pulse electromagnetic excitation. The transmitting system tends to reduce the turn-off time to enhance the induced response. There are many methods to realize fast turn-off times such as RCD circuits and voltage clamps [12,13]. According to the law of electromagnetic induction, the turn-off rate of an emission current, the change rate of a magnetic field, and the voltage of a voltage source (emf) are positively proportional. The voltage at both ends of the capacitor (C) increases continuously during the current turn-off period. Extending the turn-off time of the emission current means increasing the charging time of the capacitance and reducing the charging voltage at the same time. The energy in capacitance is the product of voltage and duration that provides charging. Turn-off time can compensate for the influence of charging voltage weakening the polarization effect. We propose that extending the turn-off time weakens the excitation of the induced field. As a result, the polarization effect can be measured more easily under this balance.

In this study, we designed the double trapezoidal wave transmitting system with a controllable turn-off edge for the induced polarization effect in TEM measurement. The system can change the turn-off time by changing the clamp voltage during the turn-off process of the emission current. The emission current with short and long turn-off times contributes to inducing the induction and polarization fields, respectively. We propose using a polarized wire loop based on the filament model to simulate the polarized earth for experiment. The effect of the system on the induced polarization effect measurement is qualitatively verified.

## 2. Theoretical Model of Turn-Off Edge Based on Passive Clamping

The turn-off time of emission current depends on the voltage at both ends during the turn-off process. The turn-off time decreases rapidly as voltage increases [14]. Turn-off time is commonly increased by high clamping voltage. A TVS diode is a typical passive clamping device. When the voltage at both ends of the TVS diode is higher than the clamp voltage, the diode breaks down instantaneously and keeps the voltage at the clamp voltage. The clamping process based on a TVS diode is shown

in Figure 2. The transmitting coil is equivalent to a circuit composed of inductance (Lc), resistance (Rc), and stray capacitance (Cs) [15]. $i(t)$ is the current flowing through the transmitting coil and $U_c$ is the voltage at both ends of the coil.

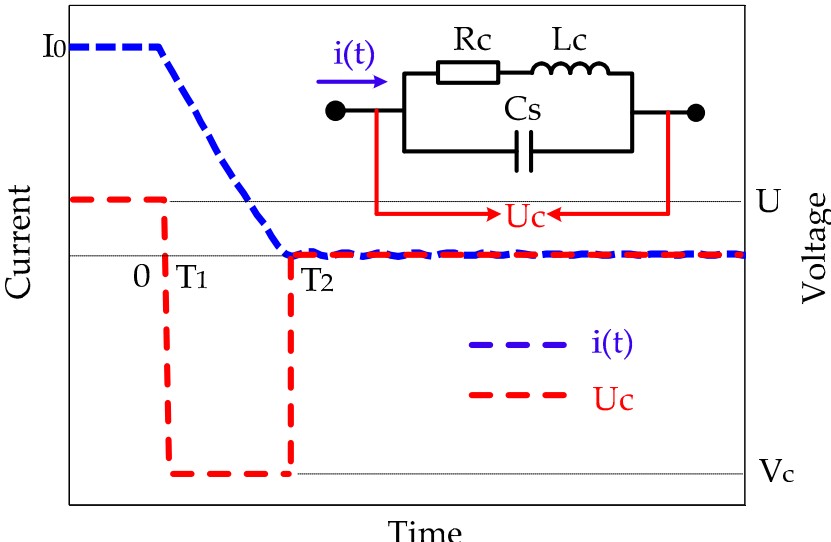

**Figure 2.** The clamping process based on a TVS diode.

The battery supply voltage of the transmitting bridge circuit is $U$. Set the turn-off time of the transmitting bridge circuit as zero time, and the current through the coil when $t < 0$.

$$i(t) = I_0 = \frac{U}{R_c}(t < 0) \tag{1}$$

Stray capacitance inevitably exists in the transmitting coil. When the emission current is turned off, the current flowing through the coil drops sharply, and the circuit composed of resistance ($Rc$), inductance ($Lc$) and stray capacitance ($Cs$) vibrates violently, resulting in high voltage overshoot:

$$\begin{cases} R_c i(t) + L_c \frac{di(t)}{dt} = -U_c(t) \\ i(t) = C_s \frac{dU_c(t)}{dt} \end{cases} \quad 0 \le t < T_1 \tag{2}$$

where $U_c$ is the voltage across capacitor $Cs$.

At $T_1$, the overshoot voltage is greater than the breakdown voltage of the TVS diode, and the voltage at both ends of the diode remains at the clamp voltage $V_c$. The current turns off approximately linearly.

$$R_c i(t) + L_c \frac{di(t)}{dt} = -U_c(t) = -V_c \quad T_1 \le t < T_2 \tag{3}$$

When the current flowing through the transmitting coil drops to zero, the TVS diode returns to the open-circuit state. The transmitting coil forms a closed loop until the current disappears.

The overshoot voltage rises rapidly, and $T_1$ is too short to be ignored. The current $i(t)$ flowing through the coil at any time in the process of passive clamping:

$$i(t) = \frac{U}{R_c} e^{-\frac{R_c}{L_c}(t-t_1)} + \frac{V_C}{R_c}\left(e^{-\frac{R_c}{L_c}(t-t_1)} - 1\right) \tag{4}$$

Turn-off time of emission current is:

$$t_d = -\frac{L_c}{R_c} \ln \frac{V_C}{V_C + U} \tag{5}$$

The above shows that changing the clamp voltage during the current drop can change the current turn-off time.

## 3. Double Trapezoidal Wave Transmitting System

### 3.1. Overall Structure of the Transmitting System

We need the emission current with different turn-off times to induce the induced polarization field, and want the emission current to have high linearity and good quality. The overall structure of the double trapezoidal wave transmitting system with a controllable turn-off edge is shown in Figure 3.

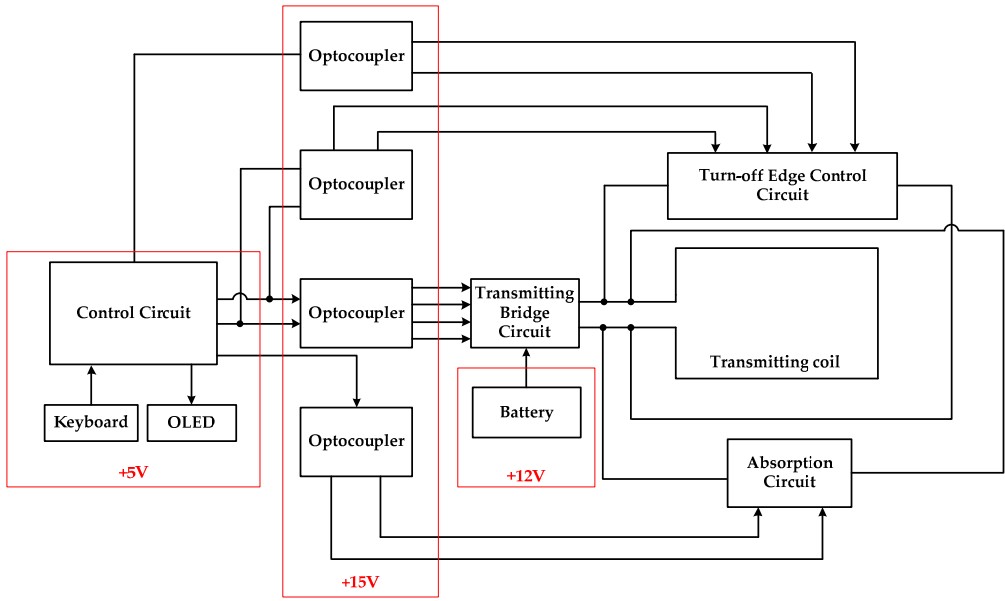

**Figure 3.** The overall structure of the double trapezoidal wave transmitting system.

The core of the control circuit is a STM32 chip based on the ARM core. Keyboard and OLED are responsible for setting parameters and display system status. All the control of the system is based on the PWM technology of the control circuit. The PWM wave is converted to 15 V by the optocoupler, which not only plays the role of level conversion but also electromagnetic isolation. The transmitting bridge is powered by a 12 V battery. Four MOSFETs that constitute the transmitting bridge are controlled by optocoupler to generate positive and negative trapezoidal waves on the transmitting coil. When the emission current is turned off, the control circuit generates signals to control the MOSFET, which constitutes the absorption circuit and the turn-off edge and can suppress the current oscillation and obtain the emission current with good quality at different turn-off times on the transmitting coil.

### 3.2. Turn-Off Edge Control Circuit

Many studies have been carried out to realize the rapid drop of emission current by high voltage clamps, and have been widely used in practical measurement [16,17]. Whereas clamp technology requires the clamping voltage to be higher than the battery voltage of the transmitting bridge, otherwise the clamp circuit will be broken down. This limits the turn-off time of the emission current. We designed the turn-off edge control circuit, which broke through the battery voltage limit. The turn-off edge control circuit is located at both ends of the coil, which can be divided into a high voltage TVS diode and a low voltage TVS diode circuit. The high voltage TVS diode clamp voltage is much higher than the battery voltage of the transmitting bridge. The transmitting bridge routes a high-power diode (D1) and four MOSFETs (S1, S2, S3, and S4). When the emission current is turned off, the transmitting coil produces a voltage overshoot with high amplitude in the same direction as the current because of the existence of inductance. Due to the diode D1, the voltage overshoot cannot

be released through the continuous-current diode of the MOSFET, which constitutes the transmitting bridge circuit. When the voltage at both ends of the transmitting coil is higher than the clamping voltage of the high voltage TVS diode, the high voltage TVS1 will be broken, and TVS1 will turn on instantaneously and keep the voltage at both ends of the transmitting coil at the clamping voltage $V_1$ of TVS1 until the current disappears completely. The coil is discharged by TVS1 in the process. Due to the high clamp voltage of $V_1$, the emission current decreases rapidly. The influence of stray capacitance is ignored in the analysis and the discharge circuit of the fast turn-off process is shown in Figure 4.

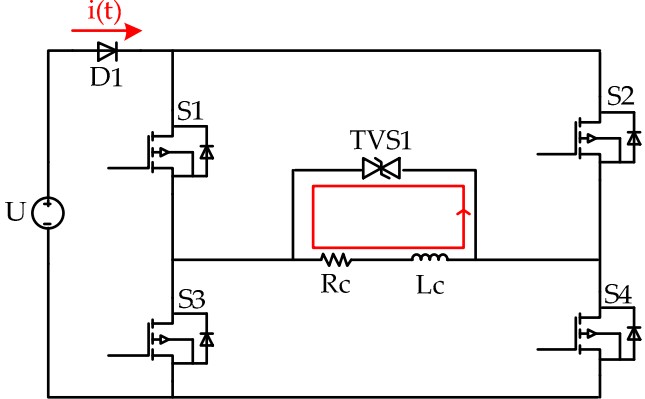

**Figure 4.** The discharge circuit of fast turn-off process.

The clamping voltage $V_1$ of TVS1 is introduced into the formula to obtain the current fast turn-off time $t_f$:

$$t_f = -\frac{L_C}{R_C} ln \frac{V_1}{V_1 + U} \tag{6}$$

The low voltage TVS diode circuit is composed of NOR gates, optocouplers, MOSFETs (S5, S6, S7, and S8), and low voltage TVS2. The low-voltage TVS diode clamp voltage $V_2$ is lower than the battery voltage of the transmitting bridge. In the current fast turn-off mode, S5/S6 are off and the low-voltage diode TVS2 does not affect the fast turn-off process.

In the slow turn-off mode, S5/S6 are on. The control signals of the transmitting bridge are the input signals of the NOR gates. When the transmitting bridge works, the outputs of the NOR gates are low and S7/S8 are off. TVS2 is isolated from the coil. However, when the current is turned off, the control signals of the transmitting bridge reach a low level and then S7/S8 are on. The NOR gates control the TVS2 to connect to both ends of the coil. Next, the coil is discharged by TVS2. The voltage at both ends of the coil is clamped on a very low voltage to realize the slow turn-off of emission current. The discharge circuit of the slow turn-off process is shown in Figure 5.

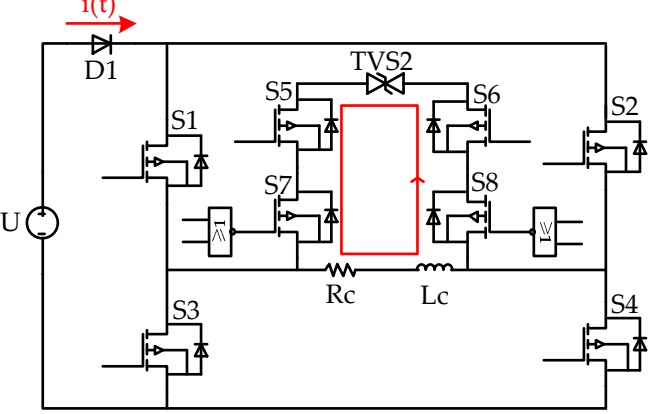

**Figure 5.** The discharge circuit of slow turn-off process.

In the process of slow turn-off of emission current, the discharge current circuit is as follows: the coil→S8→S6→low-voltage TVS diode→S5→S7→the coil or the coil→S7→S5→low-voltage TVS diode→S6→S8→the coil. The slow turn-off time $t_s$ of the emission current is:

$$t_s = -\frac{L_C}{R_C + 4R_{ON}} ln \frac{V_2}{V_2 + U} \tag{7}$$

where $R_{ON}$ is the on-resistance of MOSFET (S5, S6, S7, and S8).

### 3.3. Absorption Circuit

When the passive clamping process reaches T2, the tail of the turning-off edge will display obvious current oscillation. The analysis showed that it is caused by the characteristics of the transmitting coil. In this process, resistive ($Rc$), inductive ($Lc$), and stray capacitance ($Cs$) of the transmitting coil constitute the RLC loop. Due to the small resistance value of $Rc$, under a damped RLC, oscillation is inevitable in the loop. This will seriously affect the early signals of the secondary magnetic field [18,19]. The primary field caused by the current oscillation will also make the curve received by the sensor show a negative sign, resulting in a false polarization phenomenon. It is necessary to adopt reasonable optimization technology to reduce or eliminate the current oscillation.

Usually, the method of matching impedance R in parallel at both ends of the transmitting coil is used to suppress the oscillation phenomenon in the late turn-off period of the emission current [20]. The absorption process is:

$$\begin{cases} i(t) = C_s \frac{dU_C(t)}{dt} + \frac{U_C(t)}{R} \\ R_c i(t) + L_c \frac{di(t)}{dt} + U_C(t) = 0 \end{cases} \tag{8}$$

If the circuit is to be in a non-underdamped state and the oscillation is effectively absorbed, the matching resistance R is required:

$$R \geq \frac{R_c L_c C_s + 2L_c \sqrt{L_c C_s}}{(R_c C_s)^2 - 2L_c C_s} \tag{9}$$

However, in the current turn-off process, due to the existence of matching impedance dissipating part of the energy in the circuit, the TVS diode cannot be kept in a constant voltage clamp state for a long time. The current presents an approximately exponential decay state, which will significantly reduce the decline linearity. To avoid this situation, an absorption circuit is designed based on the characteristics of passive clamp current control. The absorption circuit consists of MOSFETs (S9/S10) and an absorption resistance R, as shown in Figure 6.

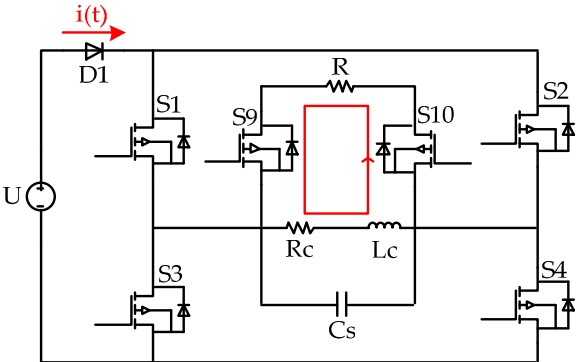

**Figure 6.** The absorption circuit.

After the current is off, during the constant voltage clamp, S9 and S10 are controlled to open accurately by the timer of the main control circuit, and the matching impedance R is cut into both ends of the transmitting coil in parallel. Because the TVS diode is in short-circuited state, the cut in

matching impedance does not affect the clamping process. Until the current is about to drop to zero, the TVS diode turns to open. The matching impedance, the coil, and the stray capacitance constitute the non-underdamped circuit, and the current begins to decline exponentially until it is zero.

## 4. Experiment with Wire Loop

### 4.1. Setup of Experiment

In geophysics, the resistance of the earth is usually equivalent by placing a closed, accurately laid out, and surveyed multi-turn, insulated ground wire loop of known inductance L and resistance R [21]. To verify the effect of the transmitting system on the polarized earth, we designed a polarized wire loop by imitating the normal wire loop. The polarized wire loop was designed according to the filament model and the multi-turn coil with known inductance was equivalent to the inductance L in the model. Next, the circuit built by selected resistance, R0, R1, and capacitance, C was connected to the two ends of multi-turn coil to form a loop. This wire loop had a resistance R0 of 400 $\Omega$, resistance R1 of 40 $\Omega$, capacitance (C) of 50 $\mu$F, and an inductance (L) of 100 mH. The equivalent circuit of the polarized wire loop is shown in Figure 7a and the physical figure is shown in Figure 7b.

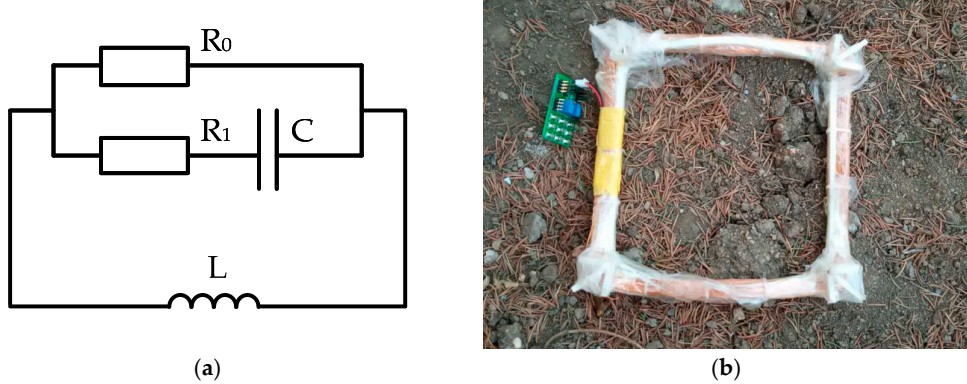

| (a) | (b) |

**Figure 7.** The polarized wire loop: (**a**) equivalent circuit and (**b**) photograph.

During the experiment, the transmitting system injected a double trapezoidal wave emission current into the 100 × 100 m square transmitting coil with a resistance of 12 $\Omega$ and an inductance of 1.05 mH, which was powered by 12 V battery. The clamping voltage of the two TVS diodes in the turning-off edge control circuit of the transmitting system was 68 and 8.2 V. We used a homemade HT SQUID for the receiver and placed it in the center of the transmitting coil. Compared with the coil receiver system, the accuracy and bandwidth of SQUID were more conducive to observing the induced polarization effect [9]. The experimental layout is shown in Figure 8. The polarized wire loop was placed coaxially and aligns the SQUID sensor to simulate the polarized earth.

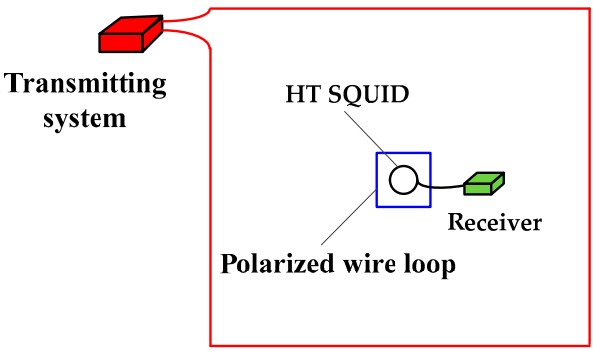

**Figure 8.** The layout of experiment.

### 4.2. Experiment and Analyses

Figure 9 shows the turning-off edge waveform of emission current observed by a clip-on ammeter. After data analysis and curve fitting, the emission current decreased from 10 A to 0 using 150 µs with a linearity of 94% at fast turn-off and using 600 µs with a linearity of 90% at slow turn-off. The tail of the current descended gently without oscillation and the waveform quality was good.

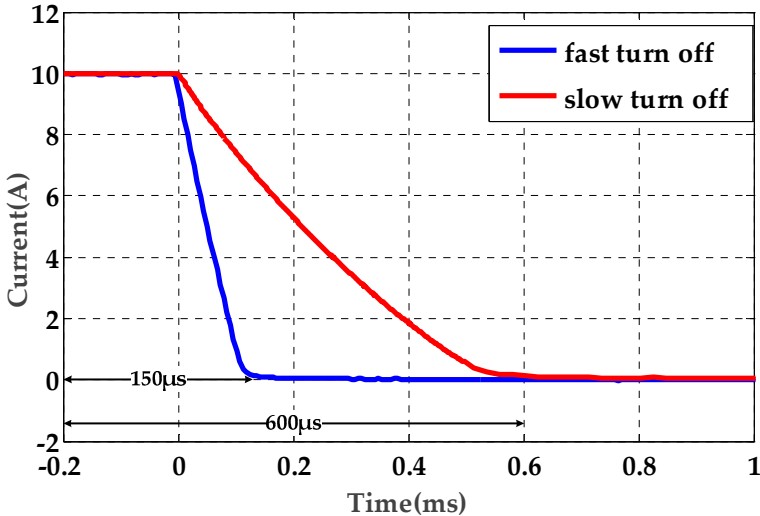

**Figure 9.** The current waveform at 150 µs turn-off time and 600 µs turn-off time.

Here, we performed two TEM measurements with and without this polarized wire loop. Figure 10 shows the response curve at a 150 µs turn-off time of the emission current. Curve I, with the wire loop, presents the sum of the background and the wire loop response. Curve II, without the wire loop, is only the background response. The first intersection of the two curves occurs at P1. The induction field of the wire loop is dominant in the early stage of the response and superimposed on the background response, so curve I is stronger before P1. With the passage of time, the polarization characteristics of the wire loop begin to appear, a negative response is superimposed on the background, and curve I decreases obviously from P1 to P2. After P2, the response gradually weakens into the receiving system noise area and the two curves tend to coincide. The two curves are consistent with the theoretical analysis, which proves that the polarized wire loop can simulate some characteristics of polarized earth.

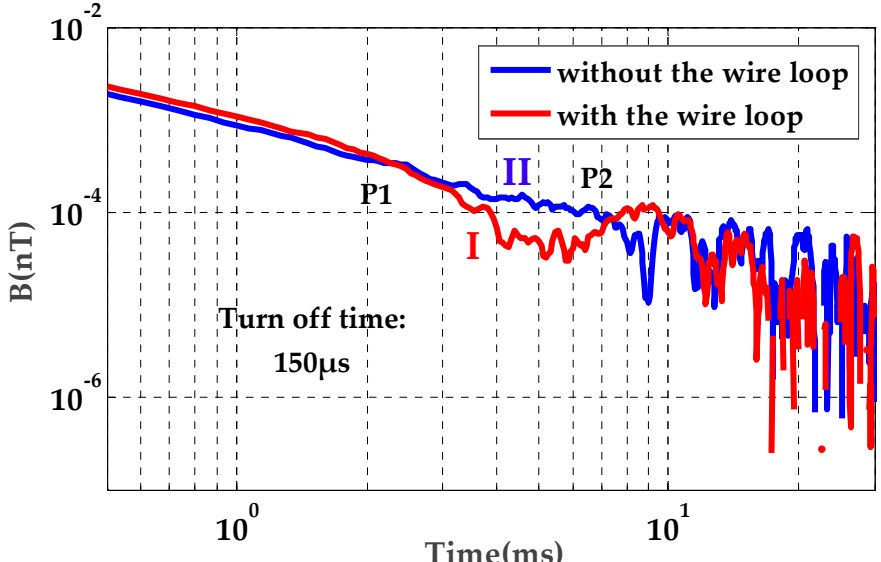

**Figure 10.** The response with and without the polarized wire loop at a 150 µs turn-off time.

Similarly, we performed two TEM measurements with and without the polarized wire loop at a 600 μs turn-off time. Curve III is the response with the wire loop and curve IV is the pure background response, as shown in Figure 11.

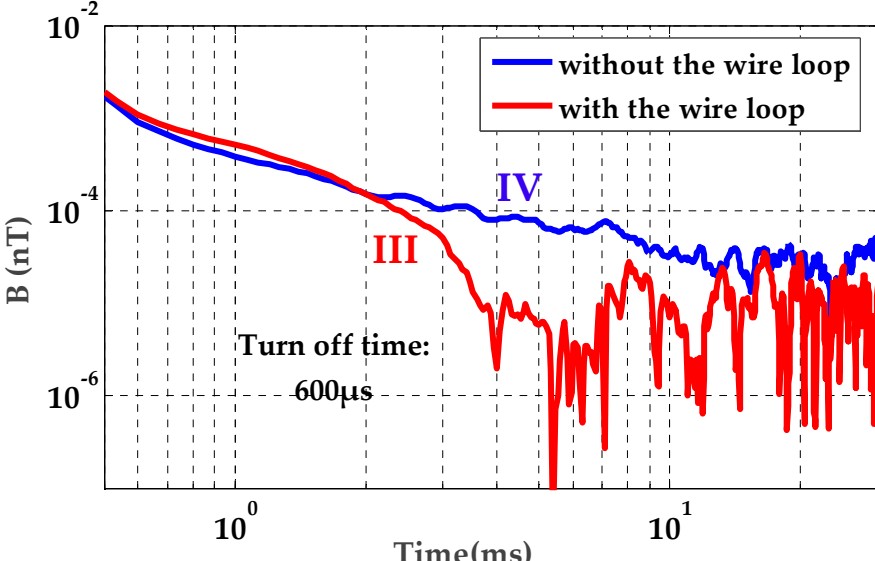

**Figure 11.** The response with and without the "polarized wire loop" at 600 μs turn-off time.

The polarized wire loop response at the 150 μs turn-off was obtained by subtracting curve II from I. The response at 600 μs turn-off time can be obtained by subtracting curve IV from III. Figure 12 shows the comparison of the wire loop responses at different turn-off times. Both curves are positive at the beginning where the induction effect is dominant. The curves then descend to zero as the polarization effect begins to appear. Finally, the curves slowly return to the positive sign and enter the system noise area. There are three obvious phenomena: (1) Before the sign becomes negative, the response intensity of curve (III − IV) is weaker than that of curve (I − II); (2) curve (III − IV) becomes negative at 2 ms, earlier than curve (I − II) by 2.3 ms; (3) the maximum negative response of the two curves is almost the same, which is $10^{-4}$ nT.

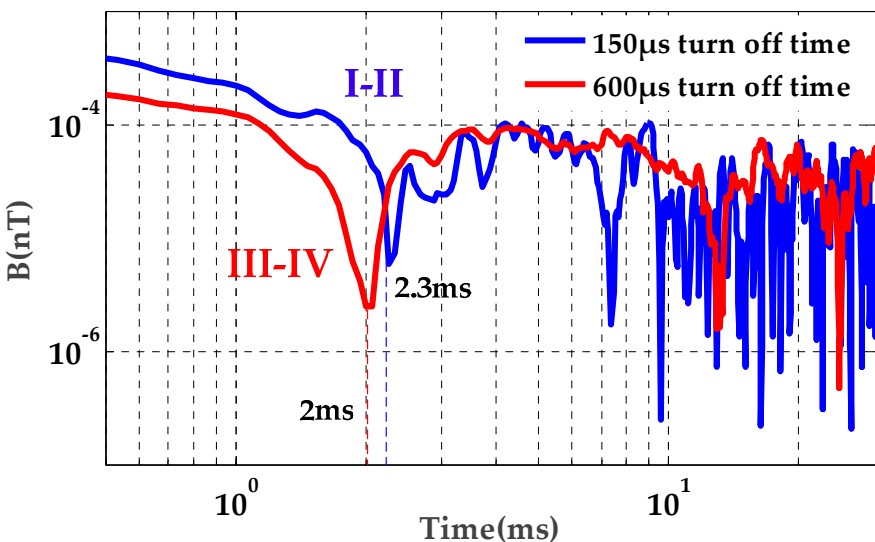

**Figure 12.** The response of the wire loop at 150 μs turn-off time and 600 μs turn-off time.

Because the polarized wire loop was designed to be used as a multi-turn coil equivalent to inductance L, resistance will inevitably be introduced into the circuit and its value cannot be ignored

compared with R1 and R0. Therefore, it is difficult to accurately extract the polarization parameters according to the filament model. We can only qualitatively analyze the influence of the transmitting system on the induced polarization effect measurement.

Combining phenomena (1) and (3), extending the turn-off time can reduce the response of the induction field, while the response of polarization field is hardly affected under the balance of the turn-off rate and duration. For the same TEM receiving system, the ratio of positive maximum to negative maximum of the response determines the measurement accuracy of polarization effect. The positive response can be amplified to the maximum range close to the receiving system, and the negative signal is amplified at the same time. Hence the response of the polarized target excited by long turn-off times can be amplified and observed more easily. The negative sign in the response is caused by the polarization effect. The earlier the negative value appears, the more polarization information the curve contains. As described in phenomenon (2), extending the turn-off time will advance the negative sign, and more polarization information can be obtained. Similarly, the short turn-off time can prolong the observation time of the induction effect. Therefore, we think that the double trapezoidal wave transmitting system with a controllable turn-off edge is suitable for the measurement of the induced polarization effect and the emission current with short and long turn-off times contributing to induce the induction and polarization fields respectively.

## 5. Conclusions

In TEM, for the measurement of the polarized earth, extending the turn-off time of the emission current means increasing the charging time of the earth's capacitance and reducing the charging voltage at the same time. We proposed that extending the turn-off time, while it can reduce the intensity of the induced field, has little effect on the pure polarization effect, which makes the polarization effect easier to observe.

In this paper, we designed a double trapezoidal wave transmitting system with a controllable turn-off edge. Firstly, by analyzing the current passive clamp process, the scheme of controlling current turn-off times by changing the clamp voltage was determined. The double trapezoidal wave transmitting system was realized by paralleling the high-voltage TVS diode at both ends of the coil and cutting in the low voltage TVS diode when the current is turned off. The current oscillation was eliminated by cutting in the absorption resistance during the current turn-off. Finally, the effect of the double trapezoidal wave emission current was qualitatively confirmed by the polarized wire loop simulated polarized earth experiment. Extending the turn-off time can weaken the response of induction field and advance the time of negative sign. The emission current with short and long turn-off times contributes to inducing the induction and polarization fields, respectively.

**Author Contributions:** Funding acquisition, Y.W.; investigation, Y.J. (Yuan Jiang); methodology, Y.J. (Yanju Ji); project administration, Y.Y., S.W., and Y.W.; supervision, Y.J. (Yanju Ji); writing—original draft, Y.J. (Yuan Jiang); Y.J. (Yuan Jiang), and Y.J. (Yanju Ji) have contributed equally to this work. All authors have read and agreed to the published version of the manuscript.

**Funding:** This research received no external funding.

**Acknowledgments:** We thank the editors and reviewers for comment.

**Conflicts of Interest:** The author declares no conflict of interest.

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
