# Peer review of "Double Trapezoidal Wave Transmitting System with Controllable Turn-Off Edge"

_applsci, doi:10.3390/app10217932_

Round 1

Reviewer 1 Report

The Manuscript entitled "Double Trapezoidal Wave Transmitting System with 2 Controllable Turn-off Edge" is associated with research in measurement the induced polarization effect based on the double trapezoidal wave transmitting system.

I would have several suggestions:
The introduction is well-writing, but it needs to be extended with modern references.

Within the entire manuscript, there are no grids at all figures in the paper. Grids need to be done.

I suggest to expand Figure 3 and at this new figure need to have all the electrical circuits of the subsystems.

Is there any experiment in the paper? At line 210 Authors says "The effect of double trapezoidal wave emission current is confirmed by the wire loop experiment based on Cole-Cole model". So, Authors should be described experiment setup and experiment results in the manuscript.

Figure 10-12 need to be clarified. Could you add information about characteristic features obtained system?

All references are very old. Could you add a modern review in the area measurement the induced polarization effect?

I suggest reconsider the manuscript after a major revision (control missing in some experiments).

Author Response

Thank you very much for your suggestions and work on this manuscript.

We have answered the questions one by one in accordance with the requirements and carefully revised the article.

Reviewer 2 Report

1) In the introduction section, it is necessary to emphasize your own contribution and novelty.

2) I think it is necessary to mention the differences and improvements of this solution compared to the solutions in references [11], [12] and [13].

3) The large thickness of the lines (connecting lines of the measured values are bold) in the graphs (Fig. 9-12) reduces the aesthetic level and readability.

4) Low graphic quality of schemes. The same type of elements should have the same size of the schematic mark (eg inductors).

5) The red lines in the circuit diagrams (Fig. 4 - 6) overlap the descriptions of the circuit elements but also reduce the readability and intelligibility of the diagrams themselves.

6) Authors cite references (eg [11-13]) that are not commonly available and their content may be in a language not used by the professional community. This does not properly assess originality, novelty and plagiarism. It would be appropriate to revise the references.

Author Response

(The authors gave the same response as above.)

Round 2

Reviewer 1 Report

The paper could be accepted at present form